# Galactomannan Pentasaccharide Produced from Copra Meal Enhances Tight Junction Integration of Epithelial Tissue through Activation of AMPK

**DOI:** 10.3390/biomedicines7040081

**Published:** 2019-10-14

**Authors:** Chatchai Nopvichai, Pawin Pongkorpsakol, Preedajit Wongkrasant, Karan Wangpaiboon, Thanapon Charoenwongpaiboon, Kazuo Ito, Chatchai Muanprasat, Rath Pichyangkura

**Affiliations:** 1Department of Biochemistry, Faculty of Science, Chulalongkorn University, Bangkok 10330, Thailand; Nopvichai.c@gmail.com (C.N.); wangpaiboon9@gmail.com (K.W.); thanapon.charoenwongpaiboon@gmail.com (T.C.); 2Department of Physiology, Faculty of Science, Mahidol University, Bangkok 10400, Thailand; pawin.pongkorpsakol1@gmail.com (P.P.); som.preedajit@gmail.com (P.W.); chatchai.mua@mahidol.ac.th (C.M.); 3Graduate School of Science, Osaka City University, Osaka 558-8585, Japan; ito@sci.osaka-cu.ac.jp; 4Division of Preclinical Sciences, Chakri Naruebodindra Medical Institute, Faculty of Medicine Ramathibodi Hospital, Mahidol University, Bangkok 10540, Thailand

**Keywords:** mannan oligosaccharide, tight junction integration, AMPK activation, Trans-epithelial electrical resistance

## Abstract

Mannan oligosaccharide (MOS) is well-known as an effective fed supplement for livestock to increase their nutrients absorption and health status. Pentasaccharide of mannan (MOS5) was reported as a molecule that possesses the ability to increase tight junction of epithelial tissue, but the structure and mechanism of action remains undetermined. In this study, the mechanism of action and structure of MOS5 were investigated. T84 cells were cultured and treated with MOS5 compared with vehicle and compound C, a 5′-adenosine monophosphate-activated protein kinase (AMPK) inhibitor. The results demonstrated that the ability of MOS5 to increase tight junction integration was inhibited in the presence of dorsomorphine (compound C). Phosphorylation level of AMPK was elevated in MOS5 treated group as determined by Western blot analysis. Determination of MOS5 structure was performed using enzymatic mapping together with ^1^H, ^13^C NMR, and 2D-NMR analysis. The results demonstrated that the structure of MOS5 is a β-(1,4)-mannotetraose with α-(1,6)-galactose attached at the second mannose unit from non-reducing end.

## 1. Introduction

Tight junction is also one of the crucial compartments of a barrier function of an epithelial tissue which is a part of anatomical barriers of innate immunity in complex and higher organisms. An impaired tight junction is a common pathologic feature in many inflammatory diseases such as inflammatory bowel disease (IBD) or Crohn’s disease or even chronic diarrhea in HIV-infected patient, or a side-effect of some drugs such as gefitinib or other drugs in EFGR-inhibitor family [1,2,3]. Impaired tight junction may lead to many symptoms such as leaking of intra- and intercellular fluid to the gut, may leading to diarrhea, malnutrition, and colitis [4,5,6,7,8]. The loss of tight junction barrier function might also lead to a dysfunction of villi that might resulting in an abnormality in nutrient absorption [1,9,10,11]. Interestingly, many recent studies had reported that β-glycans can increase tight junction formation through 5′-adenosine monophosphate-activated protein kinase (AMPK) signaling pathway which is a part of the mTOR signaling pathway [12,13,14].

Mannan oligosaccharides (MOS) is well-known as a supplementary fed for livestock which can increase the body composition, also raised immunity and stress response [15,16,17,18,19,20,21,22,23]. Although MOSs have been proven to have biological activities over the livestock quality enhancement, its mechanism of action in the tissue or cells is still unknown. Previously, MOS5, a pentasaccharide obtained from the digestion of pretreated galactomannan from copra meal with recombinant β-mannanase was reported to have the ability to enhance tight junction integration in epithelial cells [24]. Previous reports have demonstrated that β-glycan enhances tight junction integration of epithelial cells through activation of the AMPK pathway [12,13,25,26]. This infers that MOS5, which is also a β-glycan, might also enhance a tight junction integration of epithelial cells through activation of AMPK pathway.

In this study, T84 cells, a lung metastasis-colonic human colonic carcinoma cells, were used as a epithelial cell model for tight junction integrity studies induced by MOS5. Trans-epithelial electrical resistance (TEER) assay was employed. Dorsomorphine (compound C), an inhibitor of AMPK, was used to elucidate the involvement of AMPK in the activation of tight junction integration of epithelial cells by MOS5. Moreover, MOS5 structure was successfully elucidated using specific enzymatic mapping and ^1^H, ^13^C NMR, and 2D NMR analysis.

## 2. Materials and Methods

### 2.1. Cell Culture

For transepithelial electrical resistance (TEER) experiment, T84 cells (American Type Culture Collection, VA, USA) were grown in a mixture of DMEM (Invitrogen Co., Calsbad, CA, USA), supplemented with 10% *v/v* fetal bovine serum (FBS), 100 U/mL penicillin, and 100 ug/mL streptomycin. The cells were cultured in 25 cm^2^ cell culture flask (Corning life science, Tewksbury, MA, USA) maintained at 37 °C in a humidified CO_2_ incubator [12]. To form polarized monolayers, T84 cells were seeded in the Transwell^®^ insert (Corning life science, Tewksbury, MA, USA) at a density of approximately 5 × 10^5^ cells/insert and cultured for 14 days or until transepithelial electrical resistance (TEER) reached 1000 Ohm/cm^2^. The culture media were replaced daily [26].

### 2.2. Preparation of MOS5

MOS5 was produced by an enzyme hydrolytic method using RMase24 with a pre-treated galactomannan substrate obtained from copra meal in our previous study [24]. The separated MOS5 was also purified through Biogel P2 and Biogel P4 size exclusion column chromatography, respectively. The column size was 27 cm in length, 3.2 cm in diameter, with a flow rate of 0.46 mL/min at room temperature.

### 2.3. Determination of MOS5 Effects on TEER and Calcium Switch Assay

Each purified MOS5 was dissolved separately in ultrapure water to make a 50 µM to 100 µM stock concentration then filtered through 0.2-µm filter membrane before mixing with DMEM/Ham’s F-12 without FBS up to desired concentration (0.1, 1, 5, 10, and 20 µM). After the cells were grown and the polarized monolayer was formed, each prepared MOS in DMEM was treated to the cells and the change in TEER was monitored before and at 24 h after treatment. For calcium switch assay, T84 cells were cultured in DMEM in a transwell^®^ insert (Corning life science, Tewksbury, MA, USA) until cells formed the monolayer and the population of the cells reached 80% or until the TEER of the cells were steady. After that, DMEM medium was substituted with the minimum essential medium eagle, spinner modification (SMEM) (Ca^2+^-free culture media) to disrupt tight junctions. After 24 h, the SMEM medium was replaced with regular DMEM/Ham’s F-12 (containing Ca^2+^) supplemented with the vehicle, MOS5 (10 µM), MOS5 (10 µM) plus compound C (80 µM), or compound C (80 µM) alone. TEER was measured before and every 15 min after Ca^2+^ switch up to 12 h [25,26].

### 2.4. Western Blot Analysis

T84 cells were treated with 10 µM of purified MOS5, compared with non-treated group. After treatments for subjected time point, cell lysates were harvested using RIPA buffer (20 mM Tris-HCl pH7.4, 150 mM NaCl, 1 mM EDTA, 1% Triton-X100, 1% sodium deoxycholate, 0.1% SDS. Protease inhibitors: 1 mM PMSF, 5 ug/mL aprotinin and 5 ug/mL leupeptin were added prior to use). A total of 30 µg of protein was separated using sodium dodecyl sulfate polyacrylamide gel electrophoresis (SDS-PAGE) before transferring to nitrocellulose membrane. The membrane was incubated for 1 h with 5% non-fat dried milk (BioRad, Hercules, CA, USA), and incubated overnight with rabbit antibodies to phosphorylate-AMPK/Thr-172 (p-AMPK), AMPK-α and β-actin (Cell Signaling Technology, Boston, MA, USA). The membrane was then washed for four times with Tris-buffered saline Tween-20 (TBST) and incubated for 1 h at room temperature with horseradish peroxidase-conjugated goat antibody to rabbit immunoglobulin G (Cell Signaling Technology, Boston, MA, USA) [26]. The signals were detected using Luminata Crescendo Western HRP Substrate (Merck Millipore, Billerica, MA, USA). Band density was analyzed using Image J software (version 1.51s, National Institute of Health, Bethesda, MD, USA).

### 2.5. Purification of α-Galactosidase from Achantina Fulica

A crude α-galactosidase (*A. fulica*) was kindly offered by from Amano Enzyme Inc., Nagoya, Aichi, Japan and labelled as C*Af*GLA. One hundred milligrams of crude enzyme powder was diluted and mixed well with 20 mL of deionized water and precipitated with 30% ammonium sulfate solution for overnight at 4 °C. The supernatant was collected by centrifugation at 12,000× *g*, 10 min before repeating the precipitation by scaling up the ammonium sulfate concentration to 70% and 90%, respectively. The protein was then dialyzed with 5mM phosphate buffer pH 7.4 for 12 h, twice, to remove the excess ammonium sulfate salt. This dialyzed enzyme was labelled as pC*Af*GLA and was then separated through Sephadex G150 gel filtration column chromatography with gravity at 4 °C. The column size was 3 cm in diameter, 48 cm in length. Activities of separated fractions were investigated with mannobiose and melibiose to determine the digestion ability of the enzyme over β-1,4-mannosidic linkage and α-1,6-galactosidic linkage at 37 °C. The result was analyzed by TLC.

### 2.6. Structural Analysis of MOS5 Products

For structural analysis of MOS5, 50 mg of purified MOS5 was dissolved in ultrapure water before digesting directly with enzyme with excess amount of selected fractions of pC*Af*GLA at 37 °C for overnight that shown endo-mannosidase activity before purification through Biogel p4 size exclusion chromatography and then analyzed with thin layer chromatography in butanol: acetic acid: water (3:3:2) system for 3 ascents. The molecular weight of m-2 and m-3, a digestion product of MOS5, was also confirmed with MALDI imaging mass spectrometer (Solari X, FT mass spectrometry, Bruker, Billerica, MA, USA) before analyzing their structure on a FT-NMR spectrometer (AVANCE 300, Bruker, Billerica, MA, USA) at room temperature with 3-(trimethylsilyl)-1-propanesulfonic acid sodium salt (DSS) as external standard.

### 2.7. Statistical Analysis

The ratio of phosphorylation of p-AMPK/AMPK-α over of the MOS5 treated (M) and untreated group (N) at each time point was calculated by the following equation:(1)$$\mathrm{Phosphorylation}\;\mathrm{of}\;M=\left[\frac{p-\mathrm{AMPK}\;\mathrm{of}\;M}{\left(\mathrm{beta}-\mathrm{actin}\right)}\right]÷\left[\frac{\mathrm{AMPK}-\alpha \;\mathrm{of}\;M}{\left(\mathrm{beta}-\mathrm{actin}\right)}\right]$$
(2)$$\mathrm{Phosphorylation}\;\mathrm{of}\;N=\left[\frac{p-\mathrm{AMPK}\;\mathrm{of}\;N}{\left(\mathrm{beta}-\mathrm{actin}\right)}\right]÷\left[\frac{\mathrm{AMPK}-\alpha \;\mathrm{of}\;N}{\left(\mathrm{beta}-\mathrm{actin}\right)}\right]$$
(3)$$\mathrm{Percent}\;\mathrm{relative}\;\mathrm{of}\;p-\mathrm{AMPK}/\mathrm{AMPK}-\alpha \;\mathrm{to}\;\mathrm{control}\;=\left[\frac{\mathrm{Phosphorylation}\;\mathrm{of}\;M}{\mathrm{Phosphorylation}\;\mathrm{of}\;N}\right]÷\left[\frac{\mathrm{Phosphorylation}\;\mathrm{of}\;N}{\mathrm{Phosphorylation}\;\mathrm{of}\;N}\right]$$

The significant difference between samples was determined by one-way ANOVA using GraphPad Prism 7 software (GraphPad Software Inc., La Jolla, CA, USA).

## 3. Results

### 3.1. Effects of MOS5 on Tight Junction Assembly of MOS5 via AMPK Pathway

In this study, determination of MOS5 concentration results shows that a concentration at 10 µM and 20 µM of MOS5 significantly increased TEER when compared to the vehicle group (*n* = 3–4, one-way ANOVA, *p* = 0.002 and *p* < 0.0001, respectively). The concentration below 5 µM showed no difference in TEER level (Figure 1a). TEER results under a challenging of AMPK inhibitor, compound C, showed a significant difference of TEER recovering between MOS5 treatment and vehicles. TEER value from a treatment group of MOS5 + compound C showed no difference as compared to compound C treated group (*n* = 4–5, two-way ANOVA, *p* < 0.0001) (Figure 1b). To confirm this hypothesis, Western blot analysis of p-AMPK/AMPK-α expression was done. Total protein of each sample was extracted and then the blot analysis was performed and the band intensities were measured with ImageJ software. The band intensity of p-AMPK and AMPK-α were analysed with beta-actin band intensities before calculation. The results showed that the phosphorylation of AMPK was significantly increased at 60 min after administration of 10 µM MOS5 to the cells (*n* = 3, one-way ANOVA, *p* = 0.0014) (Figure 1c,d).

### 3.2. Determination of MOS5 Structure via Enzymatic Hydrolysis Assay

The result from incomplete digestion of purified MOS5 with α-galactosidase from *A. fulica* (Amano Enzyme Inc., Nagoya, Aichi, Japan) (C*Af*GLA) showed a monosaccharide band which has a different retention distance (Rf) than mannose (g), and a band with the same Rf to mannotetraose (m-4) (Figure 2a). Interestingly, once the concentration of α-galactosidase is increased, the products revealed other oligosaccharides. There was a disaccharide band with the same Rf to mannobiose (m-2) and a trisaccharide band (m-3) with a slightly different Rf than mannotriose (Figure 2a). Digestion of MOS5 with exo-β-D-mannosidase (*A. fulica*) (Seikagaku Corporation, Shiyoda-ku, Tokyo, Japan) revealed a monosaccharide band with the same Rf to mannose and a tetrasaccharide band with a different Rf from mannotetraose standard. Interestingly, recombinant β-mannannase, RMase24, cannot perform further digestion with MOS5 (Figure 2b).

Next, further purification of pC*Af*GLA was performed through Sephadex G150 gel filtration column chromatography and 5 mL was collected in each fraction. Fraction number 35 (F35) of separated pC*Af*GLA was labelled as P*Af*GLAF35, which has endo-β-mannosidase activity. Digestion of MOS5 with a lower amount of P*Af*GLAF35 than 2 µL per 1 µL of 1 µM MOS5 produced oligosaccharides larger than MOS5. A concentration of P*Af*GLAF35 used in digestion for NMR analysis was at excess to avoid a transferase by-product (Figure 2c). From this information, we can conclude the structure of MOS5 as shown in Figure 3.

The structure of MOS5 was further confirmed by NMR and MS. Digestion products of MOS5 with P*Af*GLAF35, m-2 and m-3, were collected and purified through Biogel P2 size exclusion chromatography before submitting to mass spectrometry and NMR analysis to confirm the structure. Mass of m-2 and m-3 were analyzed and showed a peak at 365 m/z and 527m/z, which indicated the molecular weight of disaccharide and trisaccharide with sodium salt, respectively (Figure 4a,b).

^1^H and ^13^C NMR spectra of m-2 and m-3 were analyzed at 300MHz and 150 MHz in D_2_O, respectively. The structural result of m-2 shown in Figure 5. The anomeric proton of mannose was identified with a chemical shift of protons following by; C1 on β-1,4 was identified at δ 5.169 ppm. C2 approximately at δ 3.98 to 4.06 ppm, and C4 approximately at δ 3.56. to 3.61 ppm. Moreover, a long-range CH proton chemical shift was also found at δ 4.731 ppm and δ 3.96 ppm which represented proton of C1 and C4 at β-1,4-mannosidic linkage, respectively. This had been confirmed with results from 13C NMR which revealed a chemical shift of carbon at following; anomeric C1 at δ 96.53 ppm, C2 at δ 72.90 and 73.23 ppm, and non-linkage C4 at δ 69.41ppm (Figure 5a–c).

Whereas ^1^H and ^13^C NMR spectra of m-3 represent the similar chemical shift signals as m-2, other different signals had been detected. ^1^H NMR spectra of m-3 revealed a signal of C1 on β-1,4 at δ 5.217 ppm. C2 approximately at δ 4.04 to 4.13 ppm, and C4 approximately at δ 3.61 ppm, but there were others proton chemical shift at δ 3.88 ppm, δ 4.04 ppm and 5.055 ppm which were identified as a proton chemical shift of C2, C4, and anomeric proton of C1 of α-1,6 linkage of galactose, respectively. The result shown on ^13^C NMR went along with the same trend as ^1^H NMR as it showed a similar pattern of carbon chemical shift of mannose at δ 96.57ppm on anomeric C1, δ 72.13 and 73.35 ppm on C2, and non-linkage C4 at δ 69.47ppm. ^13^C NMR result of m-3 also showed additional signal at δ 101.48, 71.12, and 71.99 ppm representing a chemical shift of C1 anomeric carbon, C2, and C3 of α-1,6 galactose, respectively (Figure 6a–c). Further 2D NMR analyses supporting m-2 and m-3 structures are provided in Appendix A and Appendix A, respectively.

## 4. Discussion

From our recent study, MOS5 is the main compound in crude MOS from the enzymatic digestion that shows the ability to enhance the tight junction of epithelial cells [24]. In this study we varied the concentration of MOS5 to determine the optimal comcentration that will best promote tight junction assembly. MOS5 at 10 µM was found to be the optimal concentration to promote tight junction assembly. This concentration was then used in further experiments. Interestingly, a higher concentration of MOS5, 20 µM, shows a lower trend in promoting tight junction. This phenomena was observed earlier for β-glycan activation of tight junction [12,13,25,26,27,28].This might be a result of over activation of the cellular signaling pathway. Further studies are required.

Therefore, the detailed mechanism of action of MOS5 remains unknown. However, MOS5 may increase tight junction assembly through the activation of AMPK via its downstream pathway in the epithelial cells. This hypothesis was supported by previous reports demonstrating the ability of oligosaccharides to increase tight junction assembly through the activation of AMPK [12,26]. Substantiation of this hypothesis was drawn with the determination of tight junction assembly with MOS5 in the presence and absence of compound C, an AMPK inhibitor. TEER result shows that under inhibition of AMPK with compound C, MOS5 could no longer enhance the tight junction assembly of T84 after the destruction of cellular tight junction by Ca^2+^ removal. This result indicated that MOS5 might activate cellular tight junction via AMPK pathway. To confirm this hypothesis, Western blot analysis of AMPK phosphorylation was performed to observe the changes in phosphorylation level of AMPK after MOS5 treatment. The result revealed that treatment of MOS5 over T84 can increase phosphorylation of AMPK at 60 min post-treatment. From these results it can be concluded that MOS5 activates cellular tight junction of epithelial cells through phosphorylation of AMPK.

Several studies reported that MOS molecules obtained from copra meal consist of mannose and galactose but the order of their repeating units and the structure of the functional MOS molecules remained unknown [29,30,31,32,33]. In our study, determination of MOS5 structure was performed using enzymatic mapping together with NMR analysis methods. Digestion of MOS5 with crude α-galactosidase released galactose and a tetrasaccharide which had a similar retention distance on TLC to mannotetraose standard. Traces amount of m-2 and m-3, resulted from the contamination of endo-mannosidase within the crude enzyme. Furthermore, hydrolysis of MOS5 with exo-β-1,4-mannosidase resulted in mannose unit and tetrasaccharide product, which had a different retention distance from mannotetraose. This suggested that this tetrasaccharide obtained from this reaction was a heteromer tetramer composed of mannose and galactose. It has been reported that the presence of galactose in galactomannan polymer can limit the hydrolysis activity of exo-mannosidase [34].

To determine the order of mannose and galactose unit in MOS5 structure, MOS5 was digested with endo-β-mannosidase. Ademark, et al. reported that only mannobiose and mannotriose products were obtained from the digestion of mannopentose [35], suggesting that endo-β-mannosidase cannot perform a further digestion on disaccharide or trisaccharide which will be helpful to determine the structure of MOS5. The endo-β-mannosidase digestion resulted in disaccharide and trisaccharide as the digestion products. The digestion products were purified and molecular weight of these products were confirmed by mass spectrometry analysis. The structure of the disaccharide and trisaccharide were determined by ^13^C NMR, ^1^H NMR, and 2D NMR analysis. The results showed that the disaccharide product was β-(1,4)-mannobiose, while the trisaccharide product was β-(1,4)-mannobiose with α-(1,6)-galactose attached on the second mannose unit from non-reducing end. From the results of α-galactosidase, exo-β-1,4-mannosidase digestions, and the NMR results of the disaccharide and trisaccharide product from the endo-β-mannosidase digestion, we can conclude that MOS5 is a pentasaccharide containing mannotetraose with α-(1,6)-galactose attached to the second mannose unit from non-reducing end, showed in figure3.

## 5. Conclusions

Pentasaccharide, MOS5, from copra meal digestion has the ability to increase tight junction of epithelial tissue. Enzymatic hydrolysis of MOS5 and ^13^C NMR, ^1^H NMR, and 2D NMR analysis of MOS5 hydrolytic products demonstrated that the structure of bioactive MOS5 is a β-(1,4)-mannotetraose with α-(1,6)-galactose attached to the second mannose unit from the non-reducing end. MOS5 activate tight junction integration though the activation of AMPK pathway.

## Figures and Tables

**Figure 1 biomedicines-07-00081-f001:**
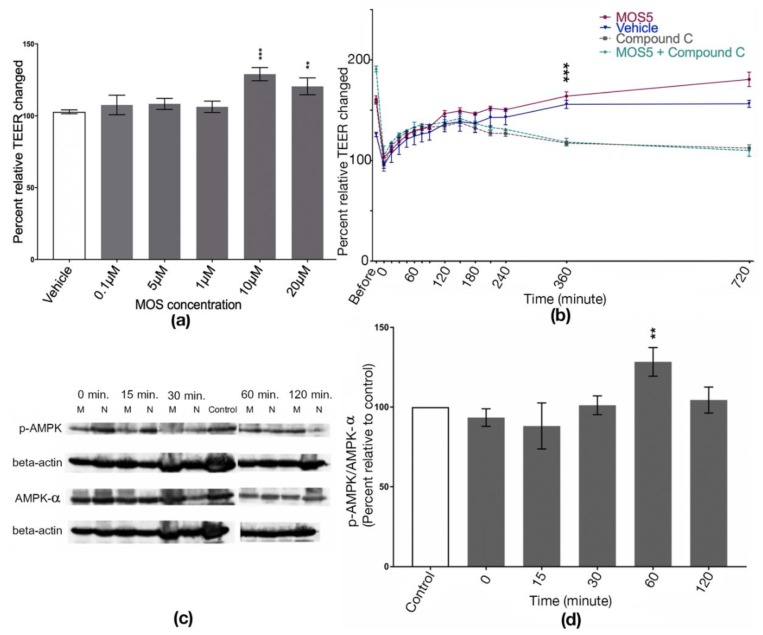
(**a**) Transepithelial electrical resistant (TEER) result of T84 cells treated with MOS5 with various concentration compared to the control group (vehicle). TEER of T84 cells were significantly increased at 20 µM and 10 µM of MOS5 (*n* = 3–4, one-way ANOVA, *p* = 0.002 and *p* < 0.0001, respectively). (**b**) Determination of tight junction reassembly of MOS5 compared to vehicle (no-treated) group and AMPK inhibitor, compound C. Result shown that MOS5 could recovered the TEER faster compared to the control but could not recovered TEER under the presence of compound C (*n* = 4–5, two-way ANOVA, *p* < 0.0001). (**c**,**d**) Western blot analysis of AMPK phosphorylation. M and N indicated T84 cells treated with 10 µM MOS5 and untreated with MOS5, respectively. The phosphorylation of AMPK was highest after the treatment of MOS5 for 60 min (*n* = 3, one-way ANOVA, *p* = 0.0014). The results here were represented as mean ± SD. (** and *** indicated a significant as *p*
≤ 0.002 and *p* < 0.0001, respectively.)

**Figure 2 biomedicines-07-00081-f002:**
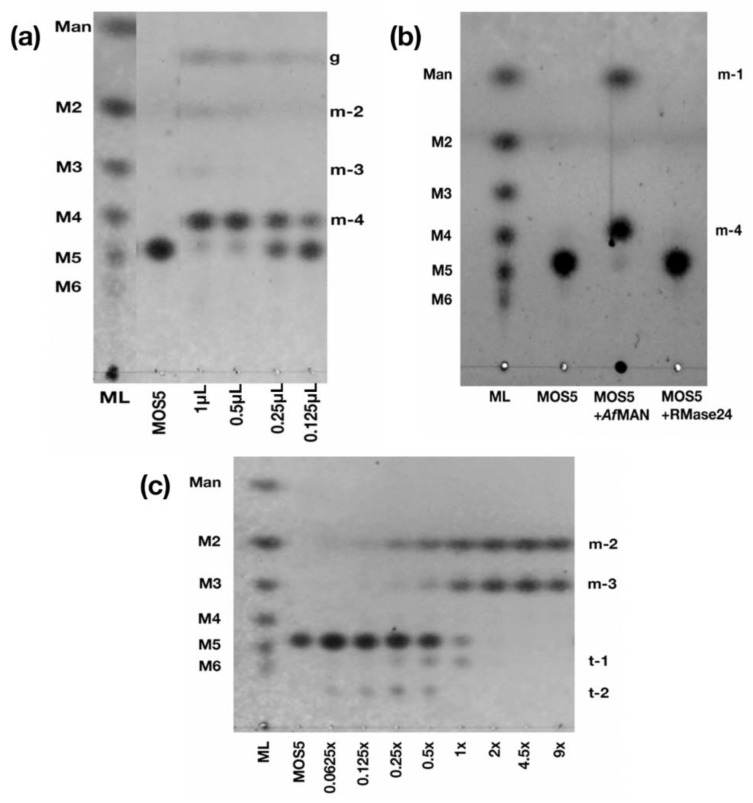
Enzymatic mapping of MOS5. (**a**) Digestion of MOS5 with C*Af*GLA. 1–0.125 µL indicates a reaction between 1 µL of 1 µM MOS5 and 1 µL to 0.125 µL of 20 mg/mL C*Af*GLA, respectively. (**b**) Digestion of MOS5 with exo-beta-D-mannosidase (*A. fulica*) (Seikagaku corporation, Japan) and RMase24. (**c**) Digestion of MOS5 with various amount of p*Af*GLAF35. 0.0625–9.0000× indicates a reaction between 1 µL of 1 µM MOS5 with 0.0625 µL to 9 µL of p*Af*GLAF35, amount ranging as labelled respectively.

**Figure 3 biomedicines-07-00081-f003:**
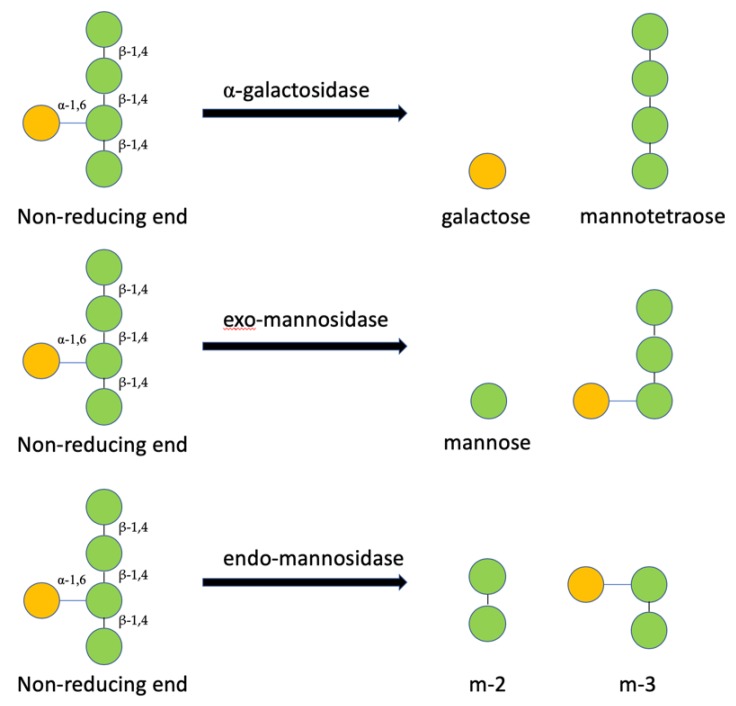
The structure of MOS5 predicted from enzymatic analysis.

**Figure 4 biomedicines-07-00081-f004:**
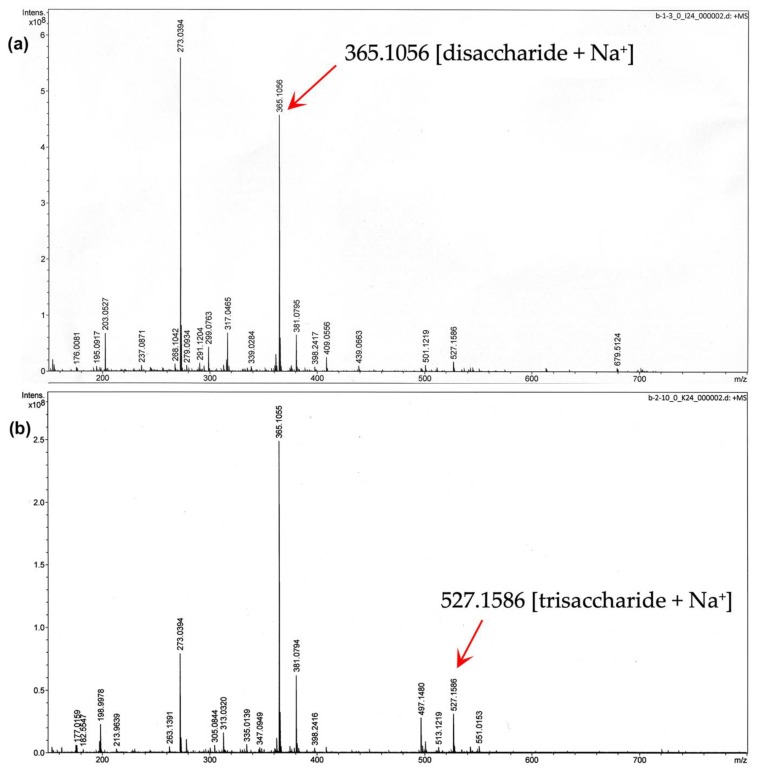
Mass spectrometry analysis of digested MOS5 (**a**) m-2 with sodium salt at 365 m/z, and (**b**) m-3 with sodium salt at 527.1586 *m*/*z*.

**Figure 5 biomedicines-07-00081-f005:**
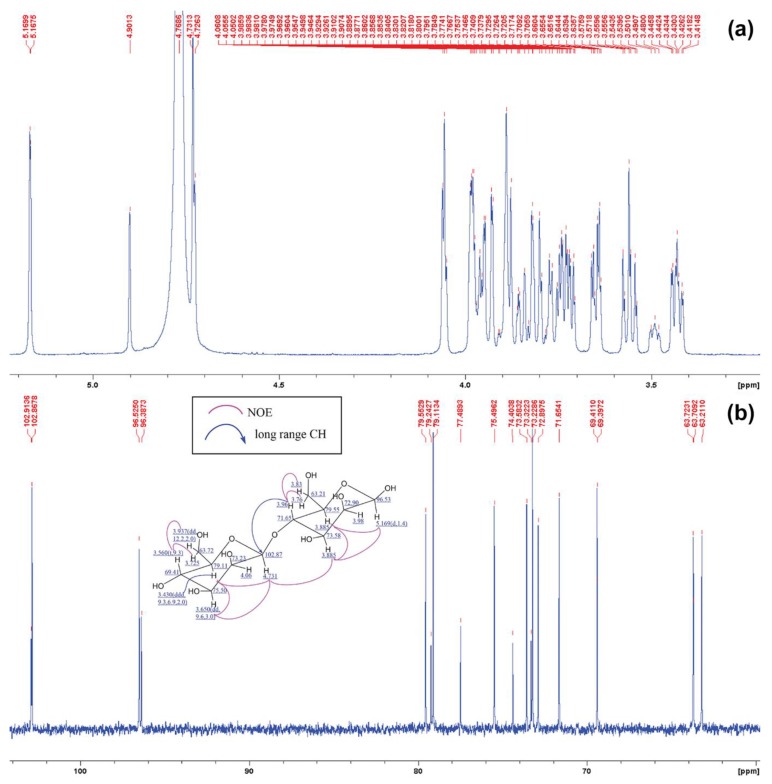
NMR analysis of P*Af*GLAF35 hydrolytic product, m-2. (**a**) ^1^H NMR of m-2 and (**b**) ^13^C NMR spectra of m-2. Predicted structure of m-2 is provided within Figure 5b.

**Figure 6 biomedicines-07-00081-f006:**
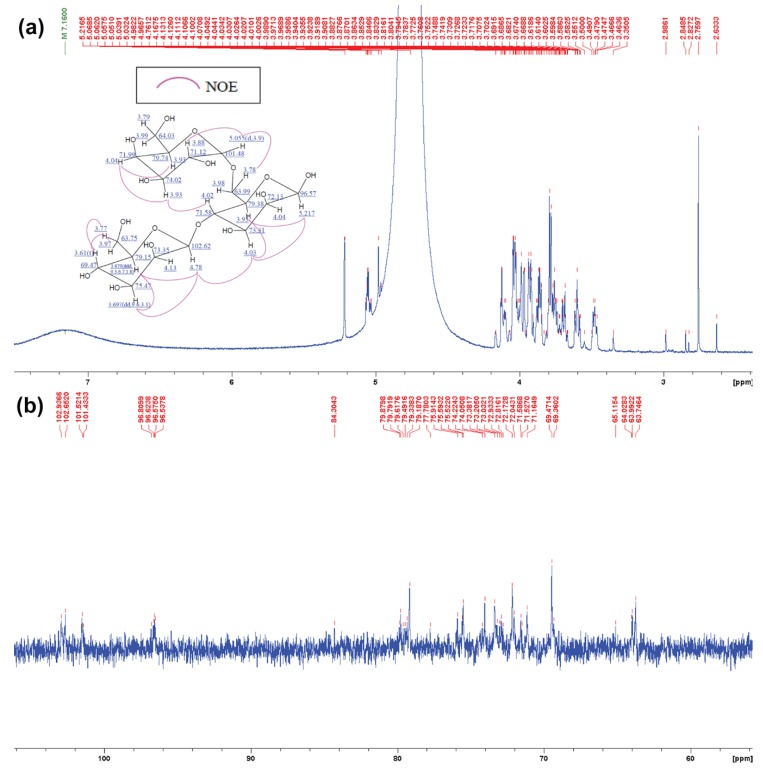
NMR analysis of P*Af*GLAF35 hydrolytic product, m-3. (**a**) ^1^H NMR of m-3 and (**b**) ^13^C NMR spectra of m-3. Predicted structure of m-3 is provided within Figure 6b.

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
