# Peer review of "Galactomannan Pentasaccharide Produced from Copra Meal Enhances Tight Junction Integration of Epithelial Tissue through Activation of AMPK"

_biomedicines, 2019, doi:10.3390/biomedicines7040081_

Round 1

Reviewer 1 Report

In this manuscript, Nopvichai et al. investigated the effect of mannanan oligosaccharide (MOS) on the tight junction of epithelial cells. They also demonstrated AMPK as a mediator of MOS action. Of importance, the authors provided new insights into the structure of MOS. Their overall findings are of interest and importance.

Suggestions:

-Materials and methods:

  The author used 80 mM of Compound C for AMPK inhibition, which is way too high. Should it be 80 μM or nM? (Liu X et al., Mol Cancer Ther 2014;13:596-605).

-Fig. 1a. High concentrations of MOS, e.g. 50 μM and 100 μM, are recommended to determine if at 10-20 μM the peak change of TEER has been induced. The EC50 should be calculated and illustrated.

-Fig. 1c. Treatments indicated by ‘N’ and ‘M’(MOS?) were not clearly stated in the main text and figure legend. The phosphorylation of AMPK-alpha seemed more evident in ‘N’ treatment. In Fig. 1d, the quantification might need to include both ‘N’ and ‘M’ treatments.

Author Response

In this manuscript, Nopvichai et al. investigated the effect of mannanan oligosaccharide (MOS) on the tight junction of epithelial cells. They also demonstrated AMPK as a mediator of MOS action. Of importance, the authors provided new insights into the structure of MOS. Their overall findings are of interest and importance.

Suggestions:

-Materials and methods:

POINT1:            The author used 80 mM of Compound C for AMPK inhibition, which is way too high. Should it be 80 μM or nM? (Liu X et al., Mol Cancer Ther 2014;13:596-605).

RESPONSE1:   It is 80uM for compound C dose in this study, not 80mM. This correction can be confirmed with a reference from our previous study (Muanprasat C., et al., Biochem phamacol 2015;96:255-236.)

POINT2:            Fig. 1a. High concentrations of MOS, e.g. 50 μM and 100 μM, are recommended to determine if at 10-20 μM the peak change of TEER has been induced. The EC50 should be calculated and illustrated.

RESPONSE2:   From our result we observed a decreasing trend after the concentration of MOS5 was increased to 20 μM. This observation was seen before in other beta glycans, COS, that the concentration above the maximal dose would have a lower activity (Yousef M., et al. Pharmacological research 2012; 66: 66-79). Therefore, we did not increase the concentration of MOS5 above 20 μM.

POINT3:            Fig. 1c. Treatments indicated by ‘N’ and ‘M’(MOS?) were not clearly stated in the main text and figure legend. The phosphorylation of AMPK-alpha seemed more evident in ‘N’ treatment. In Fig. 1d, the quantification might need to include both ‘N’ and ‘M’ treatments.

ROSPONSE3:   The information was added. The calculation of fig 1d was added to methods 2.7. The quantitation in fig 1d. was represented as p-AMPK/beta actin over AMPK/beta actin. The intensity of the bands does not reflect the signal ratio. The quantitation of the values in fig 1d. was added into the supplemental raw data.

Reviewer 2 Report

The authors in their work entitled “Galactomannan pentasaccharide produced from copra meal enhances tight junction integration of epithelial tissue through activation of AMPK” studied the mechanism of action and structure of mannan oligosaccharide on T84 cell line. The topic is actual and the work interesting.

My notices, comments, and recommendations:

P1L12: Mannan but not mannanan. Please, correct it.

P1L17: AMPK - Please, establish the abbreviation or write the whole designation.

P1L23: Trans-epithelial electrical resistance should be among the keywords.

L23: tight Junction - No capital letter in the junction.

P1L31-32: The tight junctions were lost? Should not be more suitable damaged/modified instead of the loss.

P1L33: “Other related?” What do you mean?

P1L32: Should not be more suitable damage/destruction/modification instead of the loss?

P2L10: Please add a state and town to Invitrogen.

P2L10: Please, establish the abbreviation for FBS because later you used this abbreviation without any explanation.

P2L10-11: Penicillin is commonly in um/ml and streptomycin in ug/ml. Please, correct it.

P2L11: CO2 - 2 should be in the lower case.

P2L12: Please add a town to Corning.

P2L13: It is not clear in which volume/dish were cells cultivated.

P2L24: Did you establish the abbreviation FBS? “10%” is abundant information. "Without" mean without any presence of FBS.

P2L35: Corning should be mentioned immediately after PLATES but not after CELLS/WELL.

P2L37: Did you use a commercial or home-made lysis buffer? Please, specify it.

P3L2: Amano - add a town.

P3L4: Between 12,000 and g is not a letter X but a special character.

P3L19-20: Please, add a state and town to the mass spectrometer producer.

P3L25: i) Prism is the capital letter, ii) between Prism and 7 add space, iii) add La Jolla.

P3L28-29: It is not a description of results but the discussion. Please, remove the first sentence of the paragraph. 

P3, L36: Western blot – “W” should begin by the capital letter.

P3L39-40: "The results showed that ...". Figure 1d shows pAMPK/AMPK ratio but not their absolute values.

P4, L2-10: Are the results in Figure 1 presented as mean ± SD or SEM? Please, clarify it.

Figure 1b: It is hard to distinguish corresponding curves. Please, use colors.

P4L13: Amano, Japan - Please, add a town.

P4L18: Seikagaku ... - Please, add a town.

P8L4-P9L20: One reference only is used in a short discussion. It is necessary to discuss your finding using findings of other authors that are interested in the same/similar/closely related research. Please, rewrite the discussion and add other references.

P8L5: "From our recent study" - Please, add the corresponding reference.

Author Response

The authors in their work entitled “Galactomannan pentasaccharide produced from copra meal enhances tight junction integration of epithelial tissue through activation of AMPK” studied the mechanism of action and structure of mannan oligosaccharide on T84 cell line. The topic is actual and the work interesting.

My notices, comments, and recommendations:

POINT1:            P1L12: Mannan but not mannanan. Please, correct it.

RESPONSE1:   Corrections have been made as suggested. (P1L12)

POINT2:            P1L17: AMPK - Please, establish the abbreviation or write the whole designation.

RESPONSE2:   The whole designation was added as suggested. (P1L17-18, L37)

POINT3:            P1L23: Trans-epithelial electrical resistance should be among the keywords.

RESPONSE3:   “Trans-epithelial electrical resistance” was added into the keyword list as suggested. (P1L24-25)

POINT4:            L23: tight Junction - No capital letter in the junction.

RESPONSE4:   Changes have been made as suggested.

POINT5:            P1L31-32: The tight junctions were lost? Should not be more suitable damaged/modified instead of the loss.

RESPONSE5:   As you mentioned, It would be more precise to use the phrase “impaired tight junction”. Changes have been made (P1L30, L33).

POINT6:            P1L33: “Other related?” What do you mean?

RESPONSE6:   Other related symptoms were changed to to diarrhea, malnutrition, and colitis. Citation was added. (P1L34)

POINT7:            P1L32: Should not be more suitable damage/destruction/modification instead of the loss?

RESPONSE7:   It have been changed to “Impaired tight junction”

POINT8:            P2L10: Please add a state and town to Invitrogen.

RESPONSE8:   The town and state of Invitrogen was added as suggested. (P2L16-17)

POINT9:            P2L10: Please, establish the abbreviation for FBS because later you used this abbreviation without any explanation.

RESPONSE9:   The abbreviation for FBS have been added as suggested. (P2L17)

POINT10:          P2L10-11: Penicillin is commonly in um/ml and streptomycin in ug/ml. Please, correct it.

RESPONSE10: Changes have been made; 100 IU/mL penicillin, and 100ug/mL streptomycin as standard unit use referred to our previous study. (Muanprasat, C, et al. Biochemical pharmacology 2015, 96, 225-236.) (P2L17-18)

POINT11:          P2L11: CO2 - 2 should be in the lower case.

RESPONSE11: Corrections have been made. (P2L19)

POINT12:          P2L12: Please add a town to Corning.

RESPONSE12: The information was added; NY as town and state for Corning Glass Works. (P2L20, 37-38)

POINT13:          P2L13: It is not clear in which volume/dish were cells cultivated.

RESPONSE13: Additional information concerning cell culture flask that cells were cultivated was included as suggested. (P2 L18-19)

POINT14:          P2L24: Did you establish the abbreviation FBS? “10%” is abundant information. "Without" mean without any presence of FBS.

RESPONSE14: The information concerning FBS concentration was added “10% v/v FBS” in P2L14. Also, without 10% FBS was changed to “without FBS” in P2L29.

POINT15:          P2L35: Corning should be mentioned immediately after PLATES but not after CELLS/WELL.

RESPONSE15: I cannot find what you’ve mentioned about Corning in P2L35. I’ve added (Corning life science, Tewksbury, MA,USA) after “transwell insert” as reference manufacturer in P2L33.

POINT16:          P2L37: Did you use a commercial or home-made lysis buffer? Please, specify it.

RESPONSE16: We used home-made RIPA buffer as lysis buffer. The recipe was added. (P2L47-48 to P3L1)

POINT17:          P3L2: Amano - add a town.

RESPONSE17: The prefecture of Amano enzyme industry was added. (P3L13-14)

POINT18: P3L4: Between 12,000 and g is not a letter X but a special character.

RESPONSE18: Changes was made. (P3L16)

POINT19:P3L19-20: Please, add a state and town to the mass spectrometer producer.

RESPONSE19: The information about town and state was added as suggested.

POINT20: P3L25: i) Prism is the capital letter, ii) between Prism and 7 add space, iii) add La Jolla.

RESPONSE20: Changes have been made and the information of the town was added.

POINT21:          P3L28-29: It is not a description of results but the discussion. Please, remove the first sentence of the paragraph. 

RESPONSE21: Changes have been made as suggested.

POINT22:          P3, L36: Western blot – “W” should begin by the capital letter.

RESPONSE22: Corrections have been made.

POINT23:          P3L39-40: "The results showed that ...". Figure 1d shows pAMPK/AMPK ratio but not their absolute values.

RESPONSE23: Changes have been made from “the phosphorelation of AMPK” to “the ratio of p-AMPK over AMPK- α”. The calculation methods were added. (P3 L37-45)

POINT24:          P4, L2-10: Are the results in Figure 1 presented as mean ± SD or SEM? Please, clarify it.

RESPONSE24: The results in figure 1 presented as mean ± SD. A sentence "The results were represented as mean ± SD” was added at the end of the text in Figure 1.

POINT25:          Figure 1b: It is hard to distinguish corresponding curves. Please, use colors.

RESPONSE25: Changes have been made as suggested.

POINT26:          P4L13: Amano, Japan - Please, add a town.

RESPONSE26: Changes have been made as suggested.

POINT27:          P4L18: Seikagaku ... - Please, add a town.

RESPONSE27: Changes have been made as suggested.

POINT28:          P8L4-P9L20: One reference only is used in a short discussion. It is necessary to discuss your finding using findings of other authors that are interested in the same/similar/closely related research. Please, rewrite the discussion and add other references.

RESPONSE28: Changes have been made as suggested.

POINT29:          P8L5: "From our recent study" - Please, add the corresponding reference.

RESPONSE29: Changes have been made as suggested.

Round 2

Reviewer 2 Report

Dear Authors,

I think your manuscript was highly improved.